# Experimental Study of Thermal Management Characteristics of Mass via Arrays

**Devin A. Smarra and Vamsy P. Chodavarapu \***

Department of Electrical and Computer Engineering, University of Dayton, 300 College Park, Dayton, OH 45469, USA; das_94@hotmail.com
**\*** Correspondence: vchodavarapu1@udayton.edu; Tel.: +1-937-229-2780

**Abstract:** Mass via arrays (MVAs) are intersubstrate thermal management structures that utilize thermal meta-material design principles to target localized hot spots and extreme variation in the temperature profile of electronic systems. MVAs have shown promise for integration into electronic systems as passive thermal management techniques. Theoretical analysis has shown that, when properly designed, MVAs and MVA-like structures provide control over how the heat is transferred in an electronic substrate. While the theoretical confirmation of this behavior is promising, experimental results are important to prove and strengthen MVA design principles. In this work, MVAs are implemented using an industry-standard printed circuit board (PCB) fabrication technique with Rogers 4350B (RO4350) material. The design structures are first theoretically modeled using a multilayer RO4350 stack-up in which equivalent thermal properties are studied for a variety of MVA designs. Uncertainty metrics are integrated into this model, and an iterative, Monte Carlo process is used to simulate variability in MVA performance. Next, these theoretical structures are implemented in conventional PCBs. Sample devices are chosen at random, and the heat spreading characteristics of the devices are measured using a thermal imaging camera. The results of these measurements confirm the theoretical baseline that an MVA structure provides improved thermal management characteristics relative to conventional thermal via array (TVA) structures.

**Keywords:** thermal via array; mass via array; thermal management; microsystem packaging





## 1. Introduction

Recent industry trends indicate a shift towards reducing the size of electrical systems while simultaneously increasing their computational power and functionality [1,2]. While this trend has generally been able to produce ever smaller, ever more powerful electronic systems, it has also produced devices that run hotter than ever before with high thermal densities. An obvious solution to these challenges is to reduce the heat flux density by either increasing the footprint of integrated circuits (IC) or decreasing the power consumption. Although progress has been made in developing intelligent heat sensing for localized circuit shutdown and hot spot prevention, current industry requirements and emerging high-performance applications for electric vehicles and power electronics for renewable energy require the development of new solutions [3,4].

As such, novel methods should be investigated to manage the high heat flux densities of ICs. There are two potential outlets that can be explored as a means to these ends: thermal management systems and advanced packaging concepts. Recent progress in the field of nano-structured materials has shown that nanoscopic structures, such as graphene and phase change materials, can be integrated directly into an electronic package and thermal interface materials to change how heat is transferred within a system [5–7]. Similarly, mesoscale work in electronic packaging suggests techniques, such as stacking alternating layers of dissimilar materials, texturing the surface of an object, or utilizing architected cellular structures to create artificial anisotropic properties, can result in meta-material-like behavior [8–11]. Prior research indicates that aspects of level-1 and level-2 electronic

packaging, as defined by [12], can be manipulated to change the flow of thermal energy within an electrical system, suggesting that advanced packaging concepts could prove useful in managing extremely high heat flux density sources [13].

A common modification of level-1 and level-2 aspects of electronic packaging (substrate, packaged enclosure, etc.) is the thermal via array (TVA) [14]. TVAs are through-material thermal management structures that improve heat dissipation in an otherwise thermally resistive medium. Most commonly, thermal via arrays are composed of a dense array of thermal vias—metal columns that traverse a material from thermal inlet to outlet—and planar metallization layers or heat spreaders [15,16]. The vias act as a channel in which thermal energy can be dissipated through, while the planar metallization layers provide some heat spreading capability. A great deal of time and effort has been spent trying to optimize grid spacing, grid arrangement, and various other factors of thermal via arrays [15]. While this may enable improved dissipation of heat compared to their host material, TVAs can have undesirable second-order effects. A via will dissipate a greater amount of thermal energy when compared to the surrounding medium, creating localized hotspots and, thus, substantial temperature gradients. The localized hotspot at the boundary of the vias not only creates undesirable second-order effects, but it can substantially reduce the efficiency of additional thermal management structures, such as thermoelectric coolers [17]. Furthermore, the resulting temperature gradients can cause thermal stress, which, if significantly high, can cause cracks in the substrate material [18].

An alternative through-substrate thermal management structure is the mass via array (MVA). A conventional MVA can, most simply, be thought of as a stack of N-many, independent TVAs. This structure enables a designer to leverage basic thermal metamaterial design principles to reduce the magnitude of local hot spots. The alternating layers of the thermally conductive and thermally resistive material result in a greater number of lateral thermal channels, improving the ability to spread heat while still providing an equivalent through-material heat dissipation to that of TVAs [13,19,20]. Theoretical implementation of MVAs has shown that increased planar heat spreader density can substantially improve the overall ability to spread heat from a localized source [13]. The simple nature of conventional MVA structures means that they can be implemented in both a standard printed circuit board (PCB) and multi-layer ceramic packages, such as low-temperature cofired ceramic (LTCC). This feature makes MVA structures a cost-effective, passive thermal management technique since they can be implemented during circuit board fabrication at no additional cost. While this may be the case for conventional MVAs, unconventional MVAs (uMVA) utilize strategic via placement for improving heat dissipation, making them more challenging to implement in a standard PCB.

While theoretical studies of MVAs have indicated improved thermal performance relative to TVAs, experimental verification of physical implementations of the structures has not been performed to date [13]. This work seeks to design, fabricate, and test both TVA and MVA structures using common PCB technology. The purpose of this research is to provide a theoretical framework and experimental evidence of the improved heat spreading capabilities associated with physical MVAs. This paper is organized into several sections, including System Description and Design Techniques; Design Validation; Physical Characteristics of the Test System; and Results and Discussion. The System Description and Design Technique Section reviews the design process used for the devices and motivates all subsequent steps. In the Design Validation Section, modeling and simulation are performed to establish a theoretical baseline for the MVA and TVA systems. The Physical Characteristics of the Test System Section describes the physical system, the measurement setup that is used, the MVA extraction process, and subsequent uncertainty analysis. The Results and Discussion Section presents the measured results and summarizes the findings of this research. Based on these results, this research provides preliminary performance metrics for a new type of intersubstrate thermal management structure.

## 2. System Description and Design Techniques

Prior to designing an MVA system, it must first be defined within the context of its design space. The generalized MVA architecture presented in [17] is pictured in Figure 1.

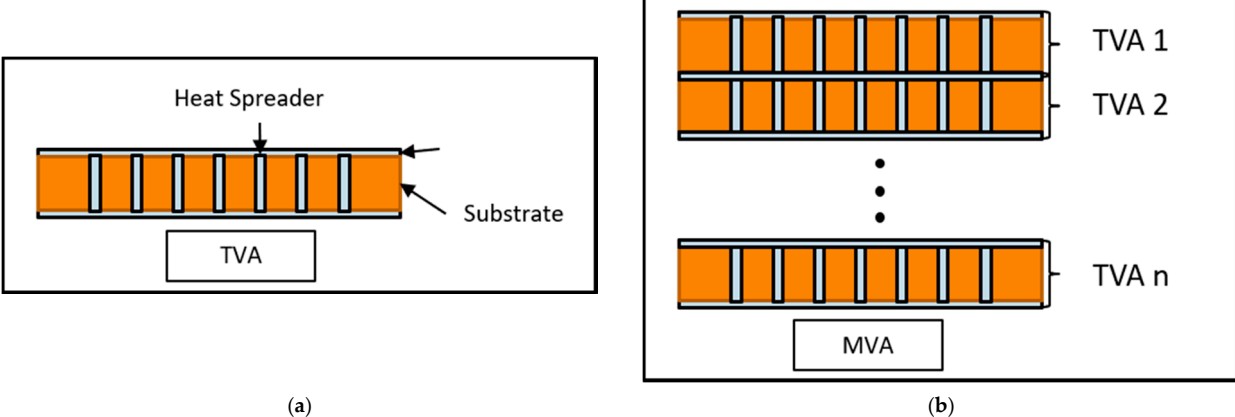

<div align="center">(a)          (b)</div>

**Figure 1.** A generalized schematic of (**a**) TVA and (**b**) MVA. As can be seen, an MVA in its most simple form is a stack of N-many TVAs.

From Figure 1, conventional MVA design space includes via density, size of vias, thickness of heat spreaders and substrate materials, heat spreader density (or the number of heat spreaders), and thermal properties of the individual constituent materials. Clearly, not all of these properties are able to be varied in a way such that sample systems can be fabricated. As such, we chose to limit the theoretical design space such that test systems can be manufactured using conventional PCB packaging systems [21].

*Full Factorial Design Format*

One technique that can be used to limit the variability of the MVA design space is the application of experimental design procedures, namely, the use of a full factorial design procedure. In addition to improving the manufacturability of these systems, this also simplifies subsequent analysis of these theoretical calculations. As such, the theoretical design space is reduced to three primary design factors—via thickness, heat spreader density, and heat spreader thickness. For convenience, these design factors are labeled X, Y, and Z, respectively. Since these devices are to be fabricated in a single run, only design factors X and Y can be implemented.

Based on this design space, we developed a 2k full factorial description of the system to greatly simplify the analysis of the calculated results. While this technique is typically reserved to analyze random experimental results, its straightforward application to results can prove to be a powerful tool to analyze the effects of, and interactions between, the various design factors. Settings for the design space are chosen such that they are within the constraints of the PCB manufacturer's fabrication model. Specifically, the high and low settings for heat spreader density are 10 and 2, respectively, and the high and low settings for via diameter are 600 μm and 200 μm, respectively. Table 1 shows the complete list of design parameters that are used by the PCB manufacturer to fabricate MVA test structures, including high and low settings of the heat spreader density and via diameter. Table 2 shows which settings are used for via diameter and heat spreader thickness in each unique MVA design.

**Table 1.** Design table for PCB MVA implementation.

| Category | Design Factor | Variable Type | Setting(S) |
|---|---|---|---|
| via | Diameter | Control | L: 200 um; H: 600 um |
| | Material | Fixed | Copper |
| heat spreader | Thickness | Fixed | 2 oz. |
| | Density | Control | L: 2; H: 10 |
| | Material | Fixed | Copper |
| substrate | Layers | Fixed | 9 |
| | Material | Fixed | RO4350 [22] |

**Table 2.** Full factorial design and notation.

| Design Factor | Symbol | X | Y | Design |
|---|---|---|---|---|
| via diameter | X | 200 μm | 2 Layers | A |
| | | 600 μm | 2 Layers | B |
| heat spreader thickness | Y | 200 μm | 10 Layers | C |
| | | 600 μm | 10 Layers | D |

## 3. Design Validation

Prior to fabrication, the thermal performance of each design is modeled using analytical techniques. These techniques are used to establish expected baselines for the heat spreading performance of each MVA design. To model this behavior, we find the through- and in-plane thermal resistance of each MVA design. These values are then converted to their dimensionless thermal conductivity values to find the thermal conductivity ratio.

### 3.1. Through-Plane Resistance

Our prior work removed the time-dependent components—heat capacity—from a generalized one-dimensional model of an MVA [19]. While a time-dependent model can be a valuable tool for determining a thermal management structure's ability to resist thermal excursions and other unpredictable thermal phenomena, those models rely only on the steady-state behavior of MVAs to determine baseline performance criteria [23,24]. While this technique is used for MVAs with solid-core vias, the same technique can be extended to an equivalent PCB-based system. Since PCB-based MVAs use cylindrical-shell vias, the generalized model proposed in [19] and used in [13] is modified to incorporate the via fill component. Figure 2 is the generalized, steady-state, one-dimensional abstraction of a PCB-based MVA. Applying basic circuit analysis techniques to the thermal circuit pictured in Figure 2, a modified expression for the through-plane thermal resistance is derived such that

$$R_{MVA} = R_{HS} + \sum_{n=1}^{n=N} R_{HS} + \left[ \frac{1}{R_{sub}} + \sum_{m=1}^{m=M} \left( \frac{1}{R_{via}} + \frac{1}{R_{via,fill}} \right) \right]^{-1} \qquad (1)$$

$$R_* = \frac{x}{Ak_*}, \qquad (2)$$

where $R_*$ is arbitrary thermal resistance, $R_{HS}$ is the thermal resistance of the heat spreader, $R_{sub}$ is the thermal resistance of the substrate, $R_{via}$ is the thermal resistance of the via, $R_{via,fill}$ is the thermal resistance of via *fill*, $x$ is the length of the path of heat transfer, $A$ is the area orthogonal to the path of heat transfer, $k_*$ is the thermal conductivity of some arbitrary material, $n$ is the number of layers in the *MVA*, and $m$ is the number of vias per layer. Using Equations (1) and (2), the through-plane thermal conductivity of the *MVA* designs shown

in Table 1 is calculated. Table 3 shows the calculated values for through-plane thermal resistance using these techniques.

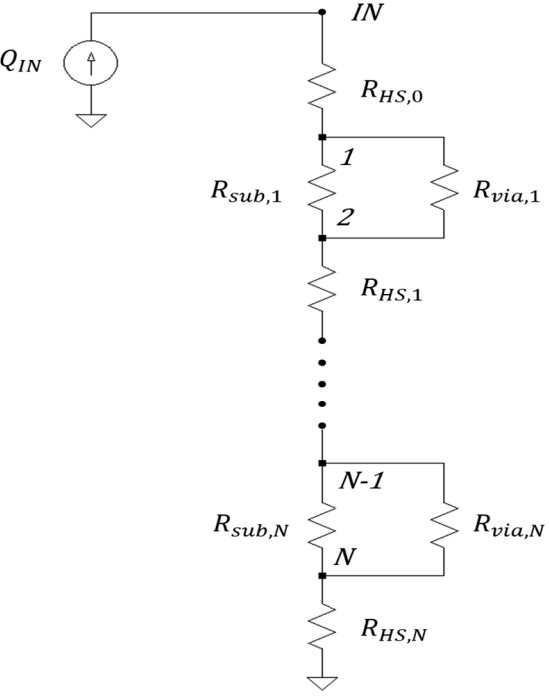

**Figure 2.** A generalized schematic representing the 1D though-plane thermal resistance of an MVA.

**Table 3.** Through-plane thermal conductivity for each MVA design.

| MVA Design | $R_{\theta,TP}$ (W/m-K) |
| --- | --- |
| A | 101.47 |
| B | 108.97 |
| C | 111.14 |
| D | 118.76 |

*3.2. In-Plane Resistance*

In-plane thermal resistance is the directional component of thermal resistance orthogonal to the direction of heat flow. Since the in-plane geometry of an MVA is substantially more complicated to calculate than the through-plane thermal resistance, we pursued a different approach to analysis. In our approach, the calculations use a unit-cell-based approach, which is a modified version of a standard finite difference approach. This description couples each via within the MVA with some amount of matrix material, proportional to the dimensions of the thermal via. This enables the thermal resistance of a unit cell to be calculated independently. Once the individual resistances of the unit cells are calculated, they can be combined with excess matrix material and conductive layers. Using the process described in our previous work, the mathematical descriptions can be developed for the MVA [13].

$$R_{ij} = \begin{bmatrix} R_{1,1} & R_{2,1} & \cdots & R_{m,1} \\ R_{1,2} & R_{2,2} & \cdots & R_{m,2} \\ \vdots & \vdots & \ddots & \vdots \\ R_{1,n} & R_{2,n} & \cdots & R_{m,n} \end{bmatrix} \tag{3}$$

$$R_{x,y} = \frac{\frac{l}{n}}{\frac{w}{m} * h * k} \tag{4}$$

where *l*, *h*, and *W* are the length, height, and width of the device, respectively. The equivalent thermal resistance of the unit cell can then be calculated such that

$$\vec{R}_i = \sum_{j=1}^{j=n} R_{ij} \tag{5}$$

$$R_{eq,UC} = \frac{1}{\sum_{i=1}^{i=m} \left[ R_i^{-1} \right]_i} \tag{6}$$

where $R_{eq,UC}$ is the equivalent thermal resistance for a unit cell. These equations are implemented into MATLAB and calculations are performed to find the in-plane thermal resistance. Table 4 shows the calculated values for in-plane thermal resistance using these techniques.

**Table 4.** In-plane thermal conductivity for each MVA design.

| MVA Design | $R_{\theta,IP}$ (W/m-K) |
|:---:|:---:|
| A | 29.04 |
| B | 32.60 |
| C | 54.95 |
| D | 61.85 |

### 3.3. Heat Spreading Validation

Work by Feng and Xu [25] has shown that the thermal conductivity ratio can be a powerful tool for determining the tendency of a structure to spread or concentrate heat. The thermal conductivity ratio is a dimensionless value that is derived from the through- and in-plane thermal resistances. According to Thompson and Ma [26], a lower value of the thermal conductivity ratio indicates improved heat spreading—as such, MVAs with a higher count of planar metallization layers should show improved heat spreading when compared to the more traditional TVA-like structures. As such, the first step in this process is to convert the values of through- and in-plane thermal resistance to their respective values of dimensional thermal conductivity. Using the dimensional thermal conductivity values, the ratio of the in-plane to through-plane thermal conductivity is calculated such that

$$K^* = \frac{K_{zz}}{K_{xx}} \tag{7}$$

where $K_{zz}$ is the thermal conductivity along the *z*-axis, and $K_{xx}$ is the thermal conductivity along the *x*-axis. Using Equation (7), the values shown in Table 5 are found for the thermal conductivity ratio for the designs in Table 1.

**Table 5.** Thermal conductivity ratio for each MVA design.

| MVA Design | K* |
|:---:|:---:|
| A | 3.49 |
| B | 3.34 |
| C | 2.02 |
| D | 1.92 |

## 4. Description of the Physical Characteristics of the Test System

To fabricate MVA test vehicles, a standard PCB manufacturing process is used to create 16 iterations of each MVA. The MVA test vehicles are designed to be compatible with several commercial, off-the-shelf radio frequency integrated circuits (RFIC). As such, the footprint of the heat spreader is 1.6 mm × 3.2 mm, and the thickness of the substrate is set to be 1.825 mm. The PCB layout of the MVA test vehicles integrates eight individual MVAs (two of each design) onto eight 40 mm × 40 mm boards.

Since the fabrication process integrates numerous MVAs into a larger PCB, individual MVAs need to be extracted so they can be measured. To accomplish this goal, hand extraction using a small grinding tool is utilized to remove each MVA; each MVA is initially rough cut from the surrounding board and then ground to size. While there are other techniques that could have been used to remove an MVA from the larger PCB, the aspect ratio—small surface area and relatively significant thickness—make milling-machine extraction challenging. Furthermore, the error caused by hand extraction provides an excellent metric for integration into an uncertainty analysis. Of the 64 original devices, only two are damaged to the point of being unusable. A picture of the extracted MVAs versus the original host PCB is shown in Figure 3.

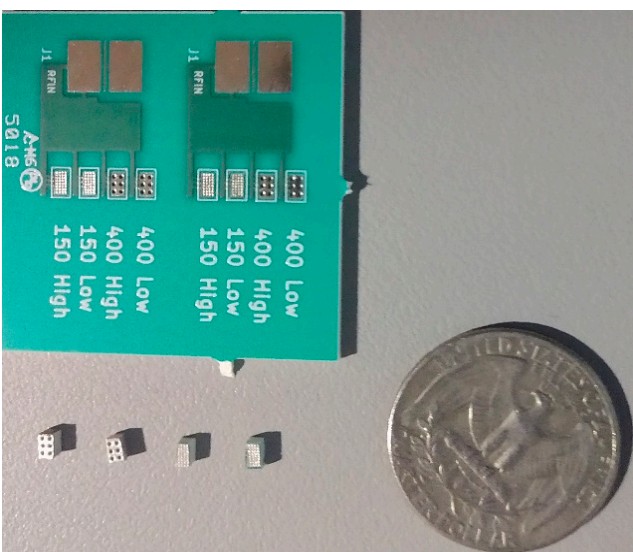

**Figure 3.** Pre- and post-extraction MVAs.

## 5. Results and Discussion

### 5.1. Extraction and Uncertainty Analysis

The width, length, thickness, and density of each device were measured and tabulated. Figures 4 and 5 are the resulting histograms of the error for each measurement. All values of error are based on a three-point average across the dimension and were calculated relative to their corresponding truth values.

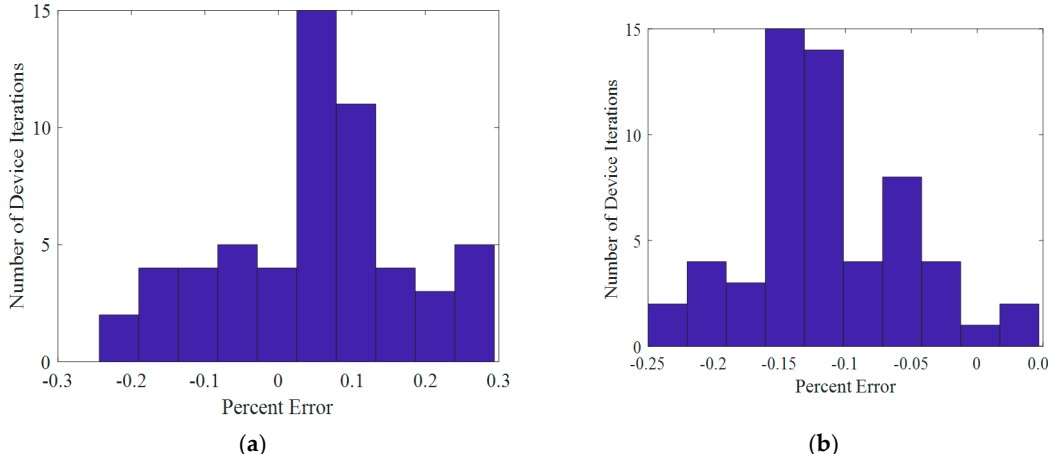

**Figure 4.** Histogram of the extraction error for: (**a**) length for all devices; (**b**) width for all devices.

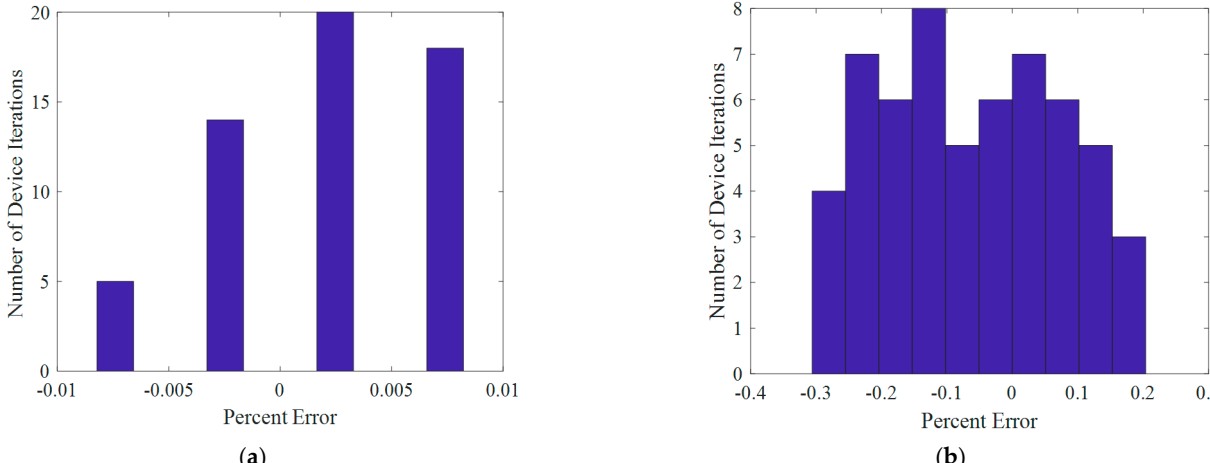

**Figure 5.** Histogram of the extraction error for: (**a**) thickness for all devices; (**b**) width for all devices.

The error results were compiled, and statistical analysis was performed on them. Figure 6 shows the box and whisker plot for each device parameter (Figure 6a) and for each MVA design (Figure 6b). The error bounds established from this statistical analysis, including the mean, maximum, minimum, and standard deviation of the measured percent error, are shown in Table 6. The statistical bounds that were calculated in the error analysis were used to create normal Gaussian distributions, which were then integrated into the thermal conductivity model described in the previous section. To perform a substantial uncertainty analysis, a Monte Carlo simulation was performed with over 1000 iterations. The resulting values are held relative to their expected truth values. Figure 7a is the resulting plot of the error-integrated value of thermal conductivity ratio for the MVA structures.

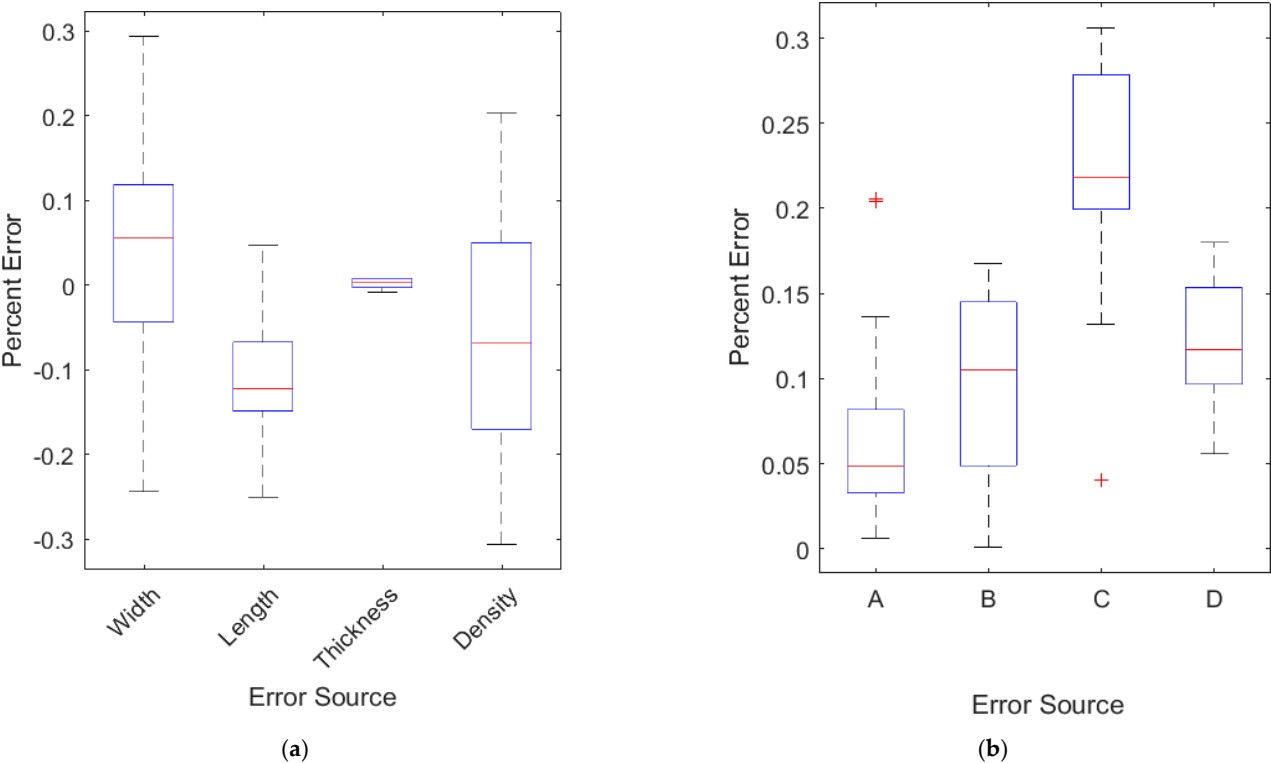

**Figure 6.** Box and whisker plot of extraction error for (**a**) source of error for all devices; (**b**) each MVA design type.

**Table 6.** Bounds calculated from MVA error measurements.

|  | $\epsilon_{W,100\%}$ | $\epsilon_{L,100\%}$ | $\epsilon_{t,100\%}$ | $\epsilon_{p,100\%}$ |
|---|---|---|---|---|
| minimum | −24.375% | −25.000% | −0.822% | −30.598% |
| maximum | 29.375% | 4.688% | 0.822% | 20.389% |
| $\sigma$ | 0.125 | 0.059 | 0.005 | 0.135 |
| $\mu$ | 5.197% | −11.447% | 0.216% | −6.133% |

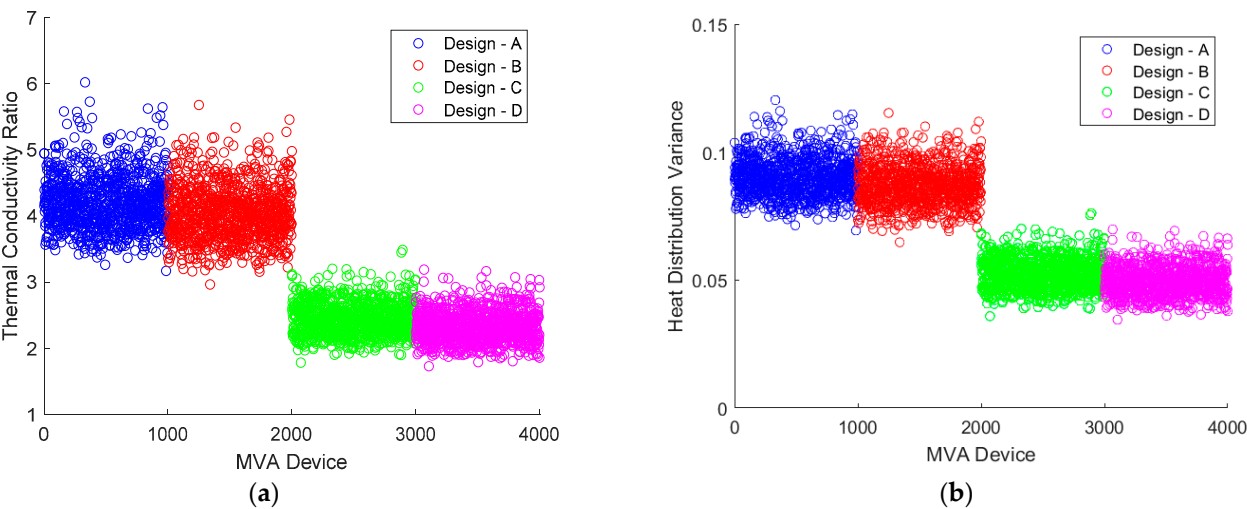

**Figure 7.** (**a**) Error-compensated theoretical thermal conductivity ratio for PCB-based MVAs; (**b**) variance of the heat spreading profile using error-compensated theoretical thermal conductivity ratio for PCB-based MVAs.

These results indicate that there can be a high degree of variance between the hand-extracted vias' thermal conductivity ratio. As such, similar variance in the measured heat spreading metrics of the fabricated MVAs can be expected. These values were integrated into the FCS model from [10,24,25] and were simulated over 101 FCS iterations. The variance of the heat distribution was calculated for each temperature profile to act as a baseline for comparison against the measured results. Figure 7b shows the modeled values of temperature profile variance based on the error-integrated calculations of thermal conductivity ratio. Table 6 shows the calculated statistical bounds for each dimension based on extracted MVA measurements.

*5.2. Description of the Measurement Setup*

To measure the heat spreading profile of PCB-based MVAs, individual MVAs were placed in a small furnace, composed entirely of ceramic foam, with a single open face. A small cavity was cut into the top surface of the furnace for the placement of an MVA test article such that the walls of the insulating material made physical contact with the boundaries of the MVA, creating pseudo-adiabatic boundary conditions on four out of six of the MVA boundaries. The top boundary of the MVA was left exposed to the ambient environment. This technique permits a system to both reach a "steady-state" heat distribution and permits the use of noncontact measurement techniques to monitor the heat spreading characteristics of the device. The bottom boundary of the MVA was heated with a low-variance heat source—created using a resistively heated thermal probe—to simulate a thermal point source. Figure 8a shows a schematic of the resulting heating apparatus.

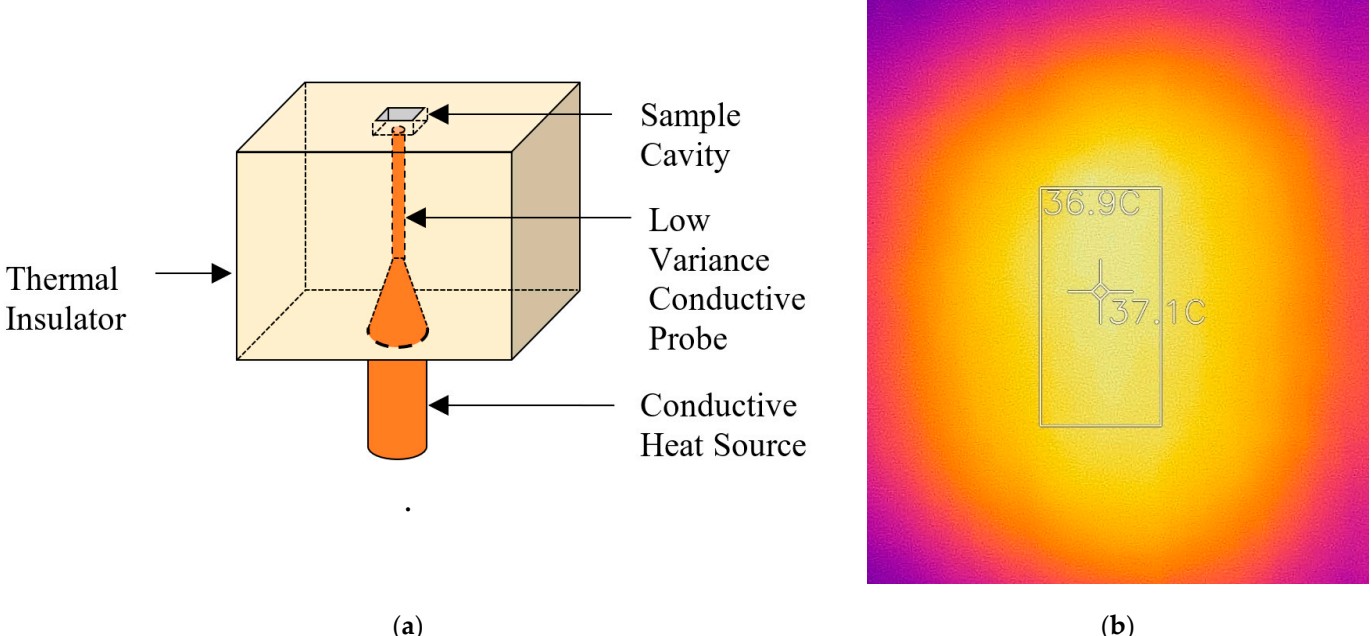

(**a**) (**b**)

**Figure 8.** (**a**) Schematic of the low-variance heat source that is used in the experiments. (**b**) Sample IR image of the surface heat distribution of an MVA device; the box indicates the position of the MVA device.

Prior to placing an individual MVA into the furnace, the system was left to heat for two hours such that it reached a thermal steady state. Once the system reached a thermal steady state, an MVA was delicately placed into the housing cut out of the furnace. A 40 μm thick Kapton film was then placed, covering the surface of the MVA and the observable region of the surrounding material; this step was performed to normalize the surface emissivity of the sample, improving the accuracy of the measurements. Once the device was placed and covered with the film, the device under test was left to reach thermal steady state,

following which an infrared (IR) image of the device was captured. Figure 8b is an example of an IR image taken of the device under test.

Post-processing was performed on each image to extract relevant information. The post-processing technique utilizes established image processing techniques, such as edge detection and thresholding, to extract the thermal profile of the MVA. The following steps were used to process the images: (1) Threshold the dynamic range of the image; (2) Adjust image bounds based on edge locations; (3) Smooth the image using a four-point smoothing filter; (4) Reshape the matrix into a n × 1 vector; (5) Perform a variance calculation across the entire heat distribution. Using this technique, information about variance was calculated and plotted for each MVA design. The results were then plotted using MATLAB. Figure 9 is a box and whisker plot of each design showing the mean and distribution of their heat spreading characteristics for all measured devices. Observation of the measured results shows that, in each case, increasing the number of planar heat spreaders improves the overall heat spreading characteristics of the device. Analysis of the plotted variance of the heat spreading performance of each of the designs indicates a 181% increase in variance between Design A and C, and a 343% increase in variance between Design B and D.

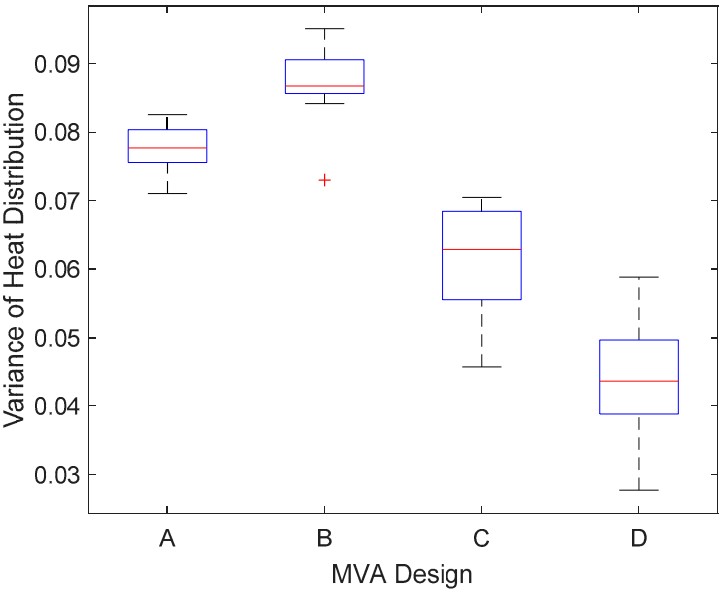

**Figure 9.** Box and whisker plot of the heat spreading characteristics for each design of MVA.

Figure 10 shows a direct comparison between the modelled and measured values for the variance of the heat distribution. As can be seen from Figure 10, these results remain entirely within the bounds established by the error-integrated results; the only exceptions to this are seen in Designs A and D, where a handful of data points lie outside of these bounds.

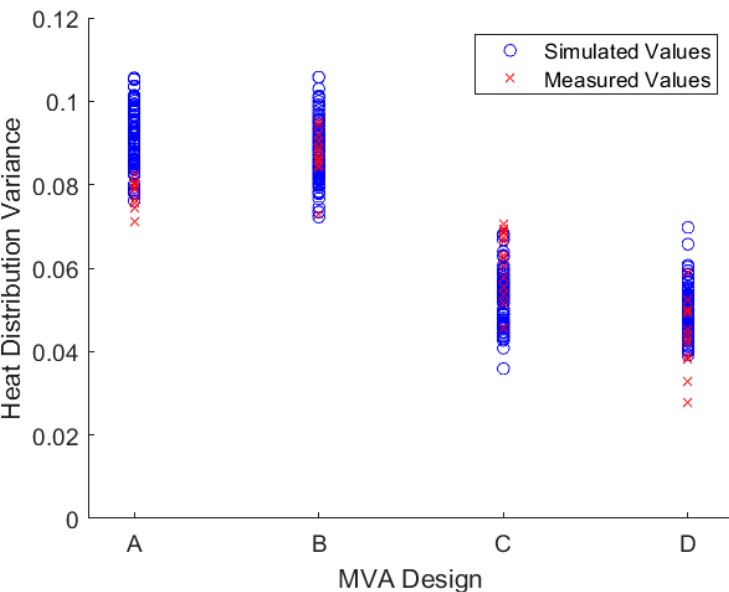

**Figure 10.** A comparison between the simulated and measured heat distribution for each design of MVA.

## 6. Conclusions

We presented an analytical model and experimental results of MVA heat spreading capabilities. Both the modeled and experimental results agree, resulting in the following conclusions: (1) our analysis confirms the observed trend that increasing the number of heat spreader layers in an MVA stack-up has a strong negative effect on the variance of the heat distribution, indicating improved heat spreading capabilities; (2) increasing the diameter of the thermal vias in an MVA has a similar, albeit weaker, effect on heat spreading capabilities; (3) there is a strong, synergistic interaction between the number of heat spreaders and the via diameter—specifically, as both the number of heat spreaders and the via diameter increase, the ability to spread heat should improve. Thus, this research has demonstrated that conventional MVAs offer improved heat spreading capabilities relative to TVAs.

**Author Contributions:** Conceptualization, D.A.S. and V.P.C.; methodology, D.A.S.; software, D.A.S.; validation, D.A.S. and V.P.C.; formal analysis, D.A.S. and V.P.C.; investigation, D.A.S.; resources, V.P.C.; data curation, D.A.S.; writing—original draft preparation, D.A.S.; writing—review and editing, D.A.S. and V.P.C.; visualization, D.A.S.; supervision, V.P.C.; project administration, V.P.C.; funding acquisition, V.P.C. All authors have read and agreed to the published version of the manuscript.

**Funding:** This research received no external funding.

**Conflicts of Interest:** The authors declare no conflict of interest.

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
