# Peer review of "Experimental Study of Thermal Management Characteristics of Mass via Arrays"

_electronics, doi:10.3390/electronics10091027_

Round 1
Reviewer 1 Report
The paper deals with Mass Via Arrays (MVA) as a passive thermal management technique. Such structures can provide effective control of heat transfer to an electronic substrate.
A theoretical model of MVAs implemented using industry-standard Printed Circuit Board (PCB) is developed and thermal properties are studied for different of MVA designs.
Some questions regarding the developed modelling methodology are the following:
1. A Full Factorial Design technique is developed to limit the variability of the MVA design space. In section 2.1 it is said that a 2k full factorial description of the system was developed to greatly simplify the analysis of results. However results in tables 1 and 2 are not clear enough with respect to the 2k designs. This section needs to be clarified to show much better how Full Factorial Design method is applied to system under study.
2. The model in section 3 is a steady state model, not including thermal capacitancies. Recent literature has introduced identification algorithm to train dynamic models of complex structures including MVA from measurements [1-2] suitable for transient thermal analysis of electronic IC designs. The authors should discuss applicability of such algorithms and discuss why they prefer to develop a steady state model. What are the implications on the design process of IC including the MVA structures as a passive thermal management technique?
3. Comparison between proposed designs A,B,C and D must be improved. It should be better explained how the different values of through- and in-plane thermal resistance determine different heat spreading capabilities for the considered designs. Why is the design B in Fig. 9 to show the highest variance of heat distribution?
4. Simulated and measured heat distribution for each design of MVA show some discrepancies, which are most evident for design A. Furthermore for designs A, B and D simulations predict higher heat distribution variance than measurements. what are the possible reasons for these deviations? How can the proposed simulation model be further improved?
[1] De Tommasi, L., Magnani, A., d'Alessandro, V., & De Magistris, M. (2014, May). Time domain identification of passive multiport RC networks with convex optimization: An application to thermal impedance macromodeling. In 2014 IEEE 18th Workshop on Signal and Power Integrity (SPI) (pp. 1-4). IEEE.
[2] De Tommasi, L., Magnani, A., & De Magistris, M. (2018). Advancements in the identification of passive RC networks for compact modeling of thermal effects in electronic devices and systems. International Journal of Numerical Modelling: Electronic Networks, Devices and Fields, 31(3), e2296.
Reviewer 2 Report
The article is worth to be proceed, but before publication it should be improved.
My suggestions are as follows:
A description of the variables used in all formulas should be placed below them.
There is no literature review from recent years. The authors cite only two publications from the last two years. Literature research should be expanded in order to be able to place the results of your research against the research of other centres.
In the introduction part, the authors refer to the literature as to group, eg [3-5], [6-9]. Please provide more details on the literature that the authors refer to.
In the final part of the introduction, it is necessary to specify what is a novelty in the presented article.
Fig 1 - please explain what substrate is used? any substrate or a specific one? It should be named.
Tabs 1, 2, 3 and 6 are insufficiently described in the text, which makes them unclear to the reader.
The measuring set is described too generally and does not allow the experiment to be repeated,
Reviewer 3 Report
The paper deals with novel methods to manage the high heat flux densities of ICs, and the investigation is at the PCB level.
1)In the abstract, the paper starts with what MVA does (shown promise for integration into electronic systems....) but what it is should be discussed first.
2)As an "industry-standard Printed Circuit Board" I would understand FR4.
This is, to the best of my knowledge, the usual PCB material.
Please comment on this in the paper along with the paper.
3)On the basis of the comments above, I suggest revising the abstract and clarify along with the paper (abstract should be informative with no much details can should be place in the core of the paper)
4) Please provide a reference/datasheet in the reference list about Rogers 15 4350B (RO4350) material to provide further details to the readers.
5) Same for RO4350 stack-up.
6) As the authors' target is reducing the "high heat flux densities of ICs", I believe the methods to face such issues at IC-level should be briefly mentioned in the introduction among the other method.
At the best of my knowledge, the first countermeasure to prevent hot spot is based on temperature sensor spread in the System-on-chip (i.e. 10.1109/ISSCC42613.2021.9366001) and/or thermal protection/shutdown circuits that locally turn off the circuit operation (i.e. 10.1109/TEMC.2011.2169964) and thus reduce locally the respective heat.
I do not see these countermeasures to react to high-temperature conditions cited/mentioned in the paper introduction. Please specifically mentioned it.
7) I believe that Fig. 1 is more clear if MVA and TVA are placed one close to each other (right/left). In Fig. 1 2 subfigure that should be visually differentiated (i.e. with a) and b) )
8)Among the comparison criteria, I do see the cost. May the authors also introduce/comment on this with a rough estimation. I believe that costs also matter.
In my view, addressing the above-mentioned doubt/comments, the paper is valuable to be published in MDPI Electronic. In fact, the model and the experimental results match, and this validates the work.
Round 2
Reviewer 1 Report
The revised version of the paper answers in a fully satisfactory manner to my comments given to the first submission.
Reviewer 2 Report
in my opinion, the article after the revision is ready to be published
Reviewer 3 Report
The authors have properly addressed my doubts/concerns.
Thanks.